# XBrainLab: An Open-Source Software for Explainable Artificial Intelligence-Based EEG Analysis

**Chia-Ying Hsieh**[1]     **Jing-Lun Chou**[1]     **Yu-Hsin Chang**[1]     **Chun-Shu Wei**[1,2,3,4]

[1]Department of Computer Science, NYCU, Taiwan
[2]Institute for Biomedical Engineering, NYCU, Taiwan
[3]Institute for Education, NYCU, Taiwan
[4]Brain Science and Technology Center, NYCU, Taiwan
{irishsieh.cs11, jalan.c, yhcgg127.cs11, wei}@nycu.edu.tw

## Abstract

Recent advancements in explainable artificial intelligence have significantly accelerated scientific discoveries across various fields. In the realm of neuroscience research, the application of deep interpretation techniques has yielded valuable insights into brain functioning and mechanisms. We introduce XBrainLab, an accessible EEG analysis tool featuring a user-friendly graphical user interface (GUI) seamlessly compatible with code scripting. XBrainLab offers a comprehensive, end-to-end deep learning EEG analysis pipeline, capable of converting raw EEG signals into comprehensible visualizations of neural patterns. Through practical demonstrations using diverse EEG datasets, we highlight XBrainLab's versatility in interpreting neural representations in alignment with established neuroscience knowledge. This evolving open-source platform bridges cutting-edge computational techniques with the frontier of neuroscientific research. The code repository can be accessed at https://github.com/CECNL/XBrainLab.

## 1  Introduction

Electroencephalogram (EEG) is a common modality for monitoring brain activity due to its non-invasiveness [1], high temporal resolution [2], and relatively low cost [3]. Despite these advantages, decoding EEG has been a technical challenge due to its low signal-to-noise ratio (SNR), contamination of artifacts, and data variability [4–6]. EEG has intrinsic limitations that stem from a low signal-to-noise ratio (SNR) and the occurrence of artifacts [4, 5], which are introduced by the lack of control in physiological mechanisms. The barrier between electrodes and the actual neural system, as well as the functional connectivity between brain regions, impairs EEG spatial resolution [7]. Additionally, the non-stationarity of neural oscillation [8, 9] makes EEG not directly understandable. To alleviate the difficulty in hand-crafting decoding algorithms for EEG, machine learning (ML) came into view for its success in automatic feature extraction and high-performance model building across various research fields. Some exemplary ML algorithms committed to EEG are: independent component analysis (ICA) [4] for blind source separation; fitting oscillations and one-over-F (FOOOF) [10] for finding periodic and aperiodic patterns in power spectral density (PSD); and filterbank common spatial pattern (FBCSP) [11] for automatically finding and selecting temporal-spatial informative EEG features.

Lately, the success of deep learning (DL) has driven the development of novel DL-based EEG processing. Starting with convolutional neural networks (CNNs) that have surpassed conventional machine learning techniques in classification [12, 13], DL models of various architectures and distinctive strengths, such as long short-term memory (LSTM) [14], transformers [15], or feature

NeurIPS 2023 AI for Science Workshop.

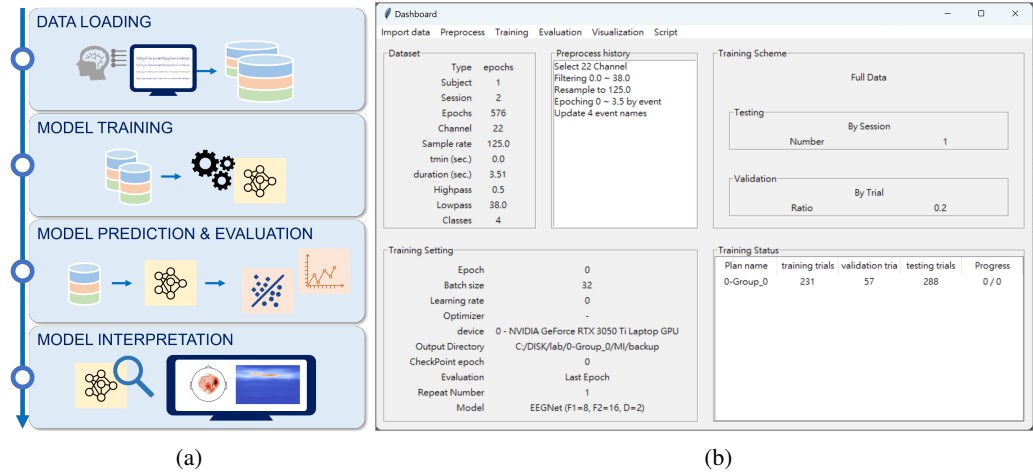

Figure 1: (a) A pipeline of EEG analysis based on XAI. (b) Dashboard of XBrainLab.

pyramid networks (FPN) [16], have elevated decoding performance in numerous EEG-related tasks. Another advancement of using DL for EEG analysis, aside from enhancing decoding performance, is the automatic feature extraction based on explainable artificial intelligence (XAI) [17, 18]. It not only reduces human labor but also avoids bias induced by prior assumptions and expert decisions while providing an objective and trustworthy deduction [12, 13].

On the other hand, incorporating machine learning (ML) into EEG analysis research requires a broad, multi-disciplinary knowledge and skills, making the development of EEG toolboxes with readily available ML algorithms of interest to the field. However, it is rare to find a tool for EEG analysis that fulfills the requirements of an open-source platform, employs cutting-edge computational techniques, and is user-friendly for users across disciplines. To address this issue, we propose the development of a novel open-source software for EEG analysis based on an XAI framework, with a focus on maximizing usability.

In this paper we present XBrainLab, an open-source software motivated to utilize XAI for EEG decoding, the pipeline and XBrainLab dashboard are shown in Fig 1b. XBrainLab is fully written in Python with an interactive graphical user interface (GUI), wrapping up EEG data processing, DL-oriented program building and model interpretation. XBrainLab sets its goal as providing a DL model-agnostic, non-exclusive dataset compatibility and out-of-the-box solution for neuroscience discovery. The remainder of this paper is organized as follows: In Section 2, we review the current situation of ML algorithms in on-shelf EEG tools, including the objective of the algorithms, building platform, and the general development direction progression over the years. Section 3 gives an overview of the XBrainLab software with descriptions of each module. In Section 4, the XBrainLab software is demonstrated with three well-studied real-life datasets to show its capability to extract different features. Finally in Section 5, we discuss the contributions and limitations of the proposed software, as well as its potential and outlook for future development.

## 2 Related Works

We survey the open-source computational neuroscience toolboxes/libraries/software embodying ML methods over the past two decades. According to the ML application on EEG, we divide tools into two categories: traditional ML and DL. The referenced standalone products for EEG and their incorporation with ML are summarized in Table1.

### 2.1 Traditional ML

**EEGLAB** EEGLAB [19] is an open-source MATLAB toolbox for single-trial EEG processing. EEGLAB came with a MATLAB script generating GUI and notably featured ICA implementation, with other ML processing methods added in later years. As one of the earliest propositions with high user-friendliness, EEGLAB is still actively maintained with a huge user base.

**BCI2000**    BCI2000 [20] is an open-source general-purpose software for brain-computer interface (BCI) research and development. BCI2000 has a GUI that is operable with shell scripts, its ML implementations cover EEG processing and classification. Although providing multiple operation modules, BCI2000 is written in C++, thus not directly compatible with popular biosignal programming languages such as MATLAB and Python.

**Brainstorm**    Brainstorm [21] is an open-source MATLAB software for MEG/EEG processing and visualization, providing script generation from GUI for advanced users and beginners. Brainstorm covers ML for EEG processing and some general ML algorithms such as support vector machine (SVM). Proposed over a decade ago, Brainstorm is still under active construction.

**Fieldtrip**    Fieldtrip [22] is an open-source MATLAB software for MEG/EEG analysis. Fieldtrip includes a large set of analysis algorithms, ranging from EEG processing, feature extraction, visualization and classification, many of which are ML methods. Prioritizing functional flexibility, Fieldtrip is designed to not providing a GUI, thus targeting users with some degree of programming experience.

**EEGsig**    EEGsig [23] is an open-source MATLAB toolbox aiming to provide a systematic EEG machine learning framework. EEGsig is inherently a GUI, with ICA for EEG processing and several machine learning classifiers for classification. Other than traditional ML classifier implementations, EEGsig outsourced MATLAB DL toolbox to provide a shallow neural network (NN). In MATLAB based environment, EEGsig is novel for being the first EEG tool accentuated on ML, and a relatively closer attempt to DL.

**MNE**    MNE-Python [24] is an open-source Python package for MEG/EEG analysis. Powered by a wide selection of scientific and numeric computing packages in Python and an active community, MNE-Python is well-documented with an enormous collection of functions. The ML methods included in MNE-Python ranges from source localization, artifact removal, and pattern recognition among others. Already widely used and established, the flaw of MNE-Python is the user threshold set by the lack of GUI support, and its genetic goal was not to accommodate DL.

## 2.2    Deep learning

**MOABB**    The mother of all BCI benchmarks (MOABB) [25] is a Python project aggregating the latest development on EEG analysis algorithms. The ML methods implemented in MOABB are mostly EEG specific, including less common implementations in other software listed in this paper, such as FBCSP [11], task-related component analysis (TRCA) [26] and Xdawn [27]. In terms of DL, MOABB provides several benchmarking convolutional neural networks (CNN). Other than the lack of GUI, since MOABB is positioned as a benchmark, the drawback lies in the extra work required for user to convert custom data into a designated format, and to learn about writing the configuration file driven pipeline.

**gumpy**    Gumpy [28] is an open-source Python toolbox for hybrid BCI, distinguished by its collection of differently structured DL models and ability to conduct analysis in a real-time paradigm. Gumpy is designed for laboratory/academic BCI research, hence providing a GUI does not seem to be in the developing consideration. During our time of literature survey, the gumpy GitHub repository appears to be out of maintenance. However, the package was used on public dataset for theoretical examination as well as real-time BCI experiments in recent studies [29, 30].

**braindecode**    Braindecode [31] is an open-source Python framework for EEG DL. Braindecode is centered around DL related functionalities, including dataset augmentation algorithms, dataset sampler and a wide selection of state-of-the-art (SOTA) DL models. Braindecode also features collecting a great deal of EEG datasets and augmentation methods, making it suitable for large-scale and reproducible SOTA DL model evaluation. With respect to DL model interpretation, braindecode provides visualization of amplitude of gradient response, although the computation and results [12] does not necessary assemble consensus XAI methods.

Overall, as DL gradually dominates the development direction of EEG decoding algorithms, the preferred environment to build toolboxes transfers from MATLAB to Python for it is the de facto

Table 1: EEG tool with ML methods. The integration refers to method implementation in software base and utilization of external code sources in provided tutorials/examples. The latter scenarios with general ML methods are marked with asterisks. CLI: command line interface.

| Name | Language | Interface | GUI Scripting | ML integration EEG specific | general | DL | DL interpr-etation |
|------|----------|-----------|-----------|--------------|---------|-----|---------------------|
| EEGLAB (2004) | MATLAB | **GUI**, CLI | **Yes** | √ | | | – |
| BCI2000 (2004) | C++ | **GUI** | **Yes** | √ | √ | | – |
| Brainstorm (2011) | MATLAB | **GUI**, CLI | **Yes** | √ | √ | | – |
| Fieldtrip (2011) | MATLAB | CLI | – | √ | √ | | – |
| EEGsig (2021) | MATLAB | **GUI** | No | √ | √ | | – |
| MNE (2013) | Python | CLI | – | √ | √* | | – |
| MOABB (2018) | Python | CLI | – | √ | √* | √ | No |
| gumpy (2018) | Python | CLI | – | | √* | √ | No |
| braindecode (2018) | Python | CLI | – | | | √ | **Yes** |
| XBrainLab (2023) | Python | **GUI**, CLI | **Yes** | | | √ | **Yes** |

language for DL. The review in this paper only includes some of the easily-found or well-used EEG toolbox with ML; for DL, the DN3 [32] and BioPyC [33] are not listed above, respective reasons are the differences in operand data representation and the ambiguity of interface definition. Although both software packages are inactive in maintenance, they are worth mentioning for being EEG-related, Python-based, and DL-incorporated.

## 3    XBrainLab: an Overview

XBrainLab is fully written in Python and distributed on GitHub under General Public License. It consists of six modules: data loading, data processing, train configuration setting, model evaluation, model visualization, and scripting. These modules form an end-to-end framework for EEG decoding with DL, as shown schematically in figure 1a. In XBrainLab, the core functions of data loading and processing are realized with MNE, a Python package widely used for EEG data preprocessing and analysis. DL-related functions are based on the PyTorch framework, and the GUI was built with Tkinter. The design and function of each module are expounded in the following subsections.

### 3.1    Data Import

EEG dataset file formats vary depending on the collecting devices, file contents, and prerequisite software. Currently the supported formats include SET, MAT, EDF/EDF+, GDF, CNT, and NPY/NPZ. Since XBrainLab aims to automatically gather necessary information from custom files, the supported formats were chosen on account of their popularity and content predictability. SET files store the primitive data structure from the EEGLAB toolbox, usually contain not only EEG data but rich information about the recording. EDF/EDF+ and GDF files store biosignal recordings with their critical information in headers or annotations, and CNT is the output format of Neuroscan systems, containing data recording and electrode locations on the scalp. MAT files are the standard format for data storage with MATLAB, which is a numerical programming language commonly used for electrophysiological processing; similarly, NPY/NPZ files come from NumPy, a fundamental Python package for scientific computing. To deal with the high flexibility in MAT and NPY/NPZ file contents, our solution in XBrainLab GUI is to allow users specify the content with a few additional clicks. Other file formats are not yet supported, and we planned to include them in XBrainLab in the near future.

### 3.2    Data Preprocessing

When analyzing EEG data, it's important to refocus on informative parts of the data by preprocessing. Several essential preprocessing methods are included in XBrainLab, and the expansion of this module will be set upon development with user feedback and advancements in preprocessing algorithms.

**Time/Window Epoching**    Sequential time series came straight out of an EEG device could be segmented based on experiment stimuli called events, and storing EEG data in cropped or uncropped representation are both possible in available datasets. For uncropped data, users can epoch them with custom event sets and time windows.

**Filtering and Resampling**    By filtering, data are restricted to a specific frequency range, forcing the following analyzing steps to focus on data parts deemed informative according to past studies. Artifacts of particular frequencies, such as low-frequency drifts or power-line noises, could also be eliminated. In addition, resampling is favored to reduce data points and save computational resources since EEG data are usually recorded with a high sampling rate.

**Event Editing and Channel Selection**    Previously, useful information and recording from all channels are automatically fetched from imported files; in XBrainLab, users can modify these information for the following analysis.

## 3.3   Dataset Splitting

EEG feature discovery could be explored in terms of experimental subjects, sessions and trials. For example, analyzing data from a single subject can extract features without cross-subject variability intervention, while analyzing data from multiple subjects extracts universal features across individuals [34]. The training/validation/test set split is provided with a graphical representation, and the loaded dataset will be split according to the custom configuration regarding subjects, sessions or trials.

## 3.4   Deep Learning

**Neural Networks**    XBrainLab implements three state-of-the-art CNN models specially designed for EEG: EEGNet [13] , SCCNet [35] and ShallowFBCSPNet [12]. The three models can serve as a deep learning baseline with empirical evidence, and for users who wish to play around with other models, the XBrainLab code structure is designed to make plugging in new models effortless.

**Settings of Training Configuration**    To reach maximum customizability in training a model, for interchangeable parameters with limited possibility, such as computing devices or available optimizers in Pytorch, will be automatically parsed for user selection. Other arbitrary parameters, such as batch size or learning rate, needs user input.

**Learning Progress Monitor**    Model's learning progress tells whether the training configuration or model selection is effective for current data besides the model performance. XBrainLab provides real-time loss and accuracy output during training, and trend curves after training is complete.

## 3.5   Result and Analysis

**Model evaluation**    After training, XBrainLab automatically computes accuracy, area under receiver operating characteristic curve(AUROC) and Cohen's kappa value, and plots the confusion matrix. Users can also export raw model output and prediction for custom metrics beyond XBrainLab.

**Post-hoc Interpretation**    In deep learning research, building models to exceed state-of-the-art performance is still one of the mainstream research motivations. The discussion on explainable artificial intelligence (XAI) [36] has been gaining more interest as the DL model behavior lacks transparency [37]. In EEG analysis with DL, the model explanations could serve as a guideline for researchers to identify meaningful parts in the signal, as the intrinsic features in EEG are not obvious upon direct inspection. In XBrainLab, the gradient response of the model on inferenced data in time domain and frequency domain, usually called saliency map [17], is provided and attainable in channel-time representation, topological plots and 3D location-time plots. With the Captum [38] package being the workhorse, exploiting more model interpretation algorithms is one of our top priority in XBrainLab development.

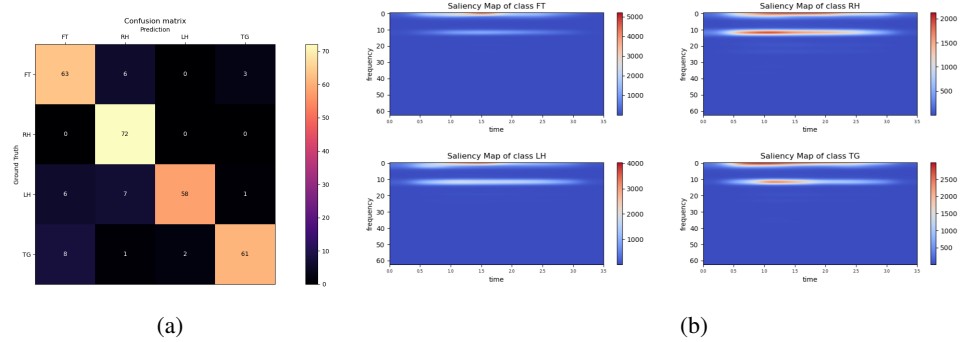

(a)                      (b)

Figure 3: Visualization of a SCCNet model in individual analysis with MI dataset subject 3. (a) Confusion matrix. (b) Saliency time-frequency spectrograms of every task. Upper left: Feet; Upper right: Right Hand; Lower left: Left Hand; Lower right: Tongue.

## 3.6 Scripting

Historical actions on XBrainLab GUI will be recorded, and the record can be exported as Python script at any time, which is executable with or without showing the GUI. This feature is designed to reduce the trouble of reproducing operations and to act as a boilerplate for custom scripts.

## 4 Case Studies

In this section we validate XBrainLab functionalities with multiple real EEG datasets. The analysis for all cases are in individual training scheme, where the trials used for training and testing are from the same subject [35]. The dataset splitting ratios follow [39]. All figures in this section are produced with XBrainLab.

### 4.1 Motor imagery

Motor imagery (MI) EEG is commonly used as BCI control, the data records body part movement intention related brain oscillations [40]. For case study, we use the BCI competition IV 2a dataset [41] for it's one of the most studied public datasets in the field [42]. The dataset was collected from 9 subjects, each subject has two sessions which is consist of 72 trials per task. There are 4 tasks

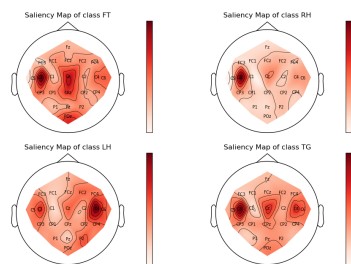

Figure 2: MI saliency topological map of a SCCNet model in individual analysis with subject 3. Upper left: Feet; Upper right: Right Hand; Lower left: Left Hand; Lower right: Tongue.

corresponding to different types of imagined movement, namely left hand, right hand, feet, and tongue. The subject selection and preprocessing procedure follows [39], only that the EEG segments in this study are extracted from 0-3.5 seconds after event cue onset.

Figure 2 and figure 4 depicts the saliency map interpretation of a trained SCCNet [35] model. For left and right hand tasks, there are strong responses in C4 and C3 channels respectively, which aligns with the literature [43] that lateral movements are corresponding to its contralateral brain region around the motor cortex. For feet and tongue tasks, the activation in middle and left of motor cortex are relatively stronger, exhibiting the model's tendency to misclassify these two tasks as each other or as the right hand. The confusion matrix of the model in analysis in shown in Figure 3a. Figure 3b further shows stronger responses from all tasks in the mu rhythm (8-12Hz), as suggested in literature [44]. Saliency map in channel-time representation and 3d location-time plots of the model in analysis can be found in the appendix.

### 4.2 Event-Related Negativity

Event-related negativity (ERN) refers to EEG perturbations after the subject encounters erroneous stimuli [13]. For case study, we use the dataset from the BCI challenge on Kaggle [45]. The dataset contains ERN responses from 26 subjects practicing P300 BCI speller task, class 0 and 1 marks

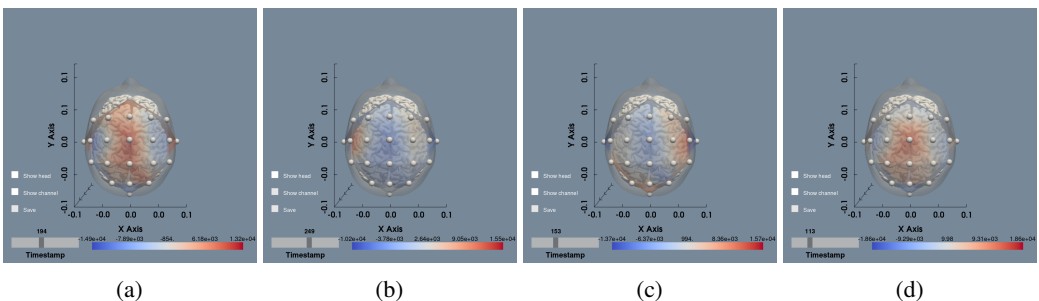

(a)     (b)     (c)     (d)

Figure 4: Saliency map of each task plotted in 3D, camera set in the horizontal plane on top of the head. (a) Feet. (b) Right hand. (c) Left hand. (d) Tongue.

correct and erroneous feedback from the speller respectively. Each subject has 5 sessions, which gives a total of 340 trials per subject. The subject selection and preprocessing procedure follows [39].

Figure 5 shows the saliency topological map interpretation across the test trials over time, and the saliency map in channel-time representation of a trained EEGNet model [13]. The topological map shows distinctive response in the FCz channel, consistent with previous findings that ERN is the most negative in the fronto-central region [46]. The pattern developed around the edge of EEG cap are presumptively muscle artifacts, relatively speaking, the color-intensive area in the parietal region could be another ERN feature manifestation [47].

In terms of channel-time representation, both classes show strong activation in the sample points about 400 500 milliseconds into the trial. For erroneous feedback, the visualization is comparable with the EEGNet model literature [13], ERN literature, [48] and SOTA DL interpretation result [39]. The confusion matrix, saliency map in 3d location-time plots and time-frequency spectrograms of the model in analysis are presented in the appendix.

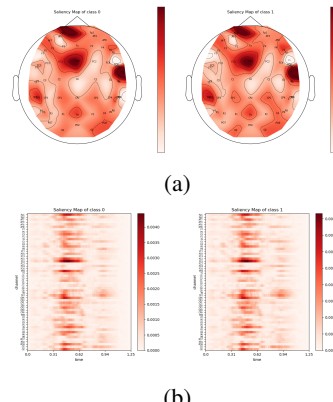

Figure 5: ERN visualization of a EEG-Net model in individual analysis with subject 7. (a) saliency topological map. (b) channel-time saliency map.

### 4.3   Steady-State Visual-Evoked Potential

Steady state visual evoked potential (SSVEP) refers to quasi-periodic oscillatory responses from the brain evoked by stimuli flickering in certain frequencies [49]. For case study we use the MAMEM SSVEP experiment 2 dataset [50]. The dataset was collected from 11 subjects. In the experiments, the subjects are instructed to focus on a box flickering in one from the five different frequencies, namely 6.6, 7.5, 8.57, 10, and 12Hz. In the original dataset, each subject has 5 sessions, and every session is consisted of 25 trials. Subject selection, channel selection, trial segmentation and preprocessing procedures follow [39].

Figure 7 shows the visualizations of a trained SCCNet model [35]. From the saliency map in channel-time representation (figure 7a), the interleaved low/high values form roughly discernible patterns of different frequencies. Furthermore, the most intense activations of each class unanimously appear in the Oz channel, which locates in the middle of the occipital lobe (figure 7c), revealing the nature of the data are visual tasks. In figure 7b the fundamental and harmonic frequencies can be identified in respective spectrograms. Another observation deduced from the confusion matrix shown in figure 6 is that some stimulus frequencies are more challenging for the model. Although the training set has slightly unequal number of trials

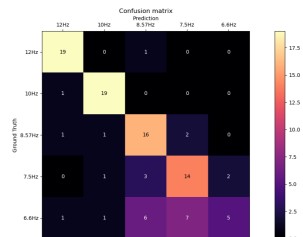

Figure 6: Confusion matrix of a SC-CNet model in individual analysis with SSVEP dataset subject 11.

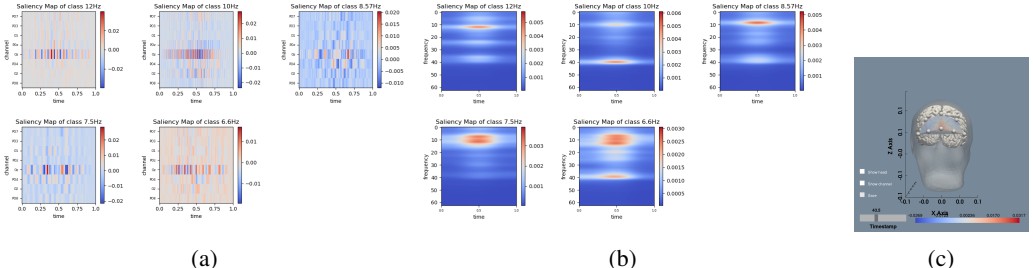

| (a) | (b) | (c) |

Figure 7: Visualization of a SCCNet model in individual analysis with SSVEP dataset subject 11.(a) Saliency map in channel-time representation. (b) Saliency time-frequency spectrograms. (c) Saliency map from a time sample, averaged across all test trials and projected on 3d mesh. Figure from class 7.5Hz is taken as example.

(4:5:6:5:5 for the 5 classes) for each task, this phenomenon is likely to be stemming from the fact that the SNR is unevenly distributed across different frequency bands, as previous studies suggested[51].

# 5    Conclusion and Future Work

We have developed XBrainLab, an open-source Python software for explainable EEG decoding. designed for explainable EEG decoding. Apart from other computational tools, XBrainLab uniquely combines critical raw EEG signal processing, deep learning, and interpretable visualization methods, all with easy access for both basic and advanced EEG analysis. Our future plans involve extending the documentation to include examples of common EEG paradigms and broadening the range of available visualization and preprocessing algorithms. In summary, XBrainLab serves as an essential tool for researchers of all levels, facilitating neuroscientific breakthroughs through advanced explainable deep learning techniques in EEG data analysis.

## Acknowledgement

This work was supported in part by the National Science and Technology Council (NSTC) under Contracts 109-2222-E-009-006-MY3, 110-2221-E-A49-130-MY2, 110-2314-B-037-061, 112-2321-B-A49-012, and 112-2222-E-A49-008-MY2; and in part by the Higher Education Sprout Project of National Yang Ming Chiao Tung University and Ministry of Education. Corresponding author: Chun-Shu Wei (wei@nycu.edu.tw).

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

# A    XBrainLab core packages

XBrainLab is completely written in Python. In this section we list the foundational packages for our software in alphabetical order with short narrations and their usage in XBrainLab. All packages are open-source.

**Captum**    Captum [38] is a model interpretability and understanding library under the PyTorch framework, implemented several widely-recognized explanation methods. The post-hoc interpretation part of XBrainLab will be depending on Captum to do the heavy lifting.

**matplotlib**    Matplotlib [52] is designed for data visualization in Python. All 2d figures in XBrainLab is plotted by matplotlib and shares its figure viewing and I/O features.

**MNE-Python**    MNE-Python [24] is a widely-used library specialized in MEG/EEG analysis. In XBrainLab, we utilize MNE for the data loading, preprocessing modules and 2d topological map visualization; the 3d human head and brain mesh objects were retrieved from MNE source code.

**NumPy**    Numpy [53] is the primary library for numeric multi dimensional array manipulation in Python. XBrainLab heavily employs NumPy for self-defined structure elements as well as intermediate data container at all scales.

**PyVista**    PyVista [54] is a Python translation of the open-source, C++ based Visualization Toolkit (VTK) software. PyVista is bound with 3d topological map visualization in XBrainLab, such as the construction of interactive window, 3d objects display and mesh generation.

**PyTorch**    Pytorch [55] is a deep-learning framework adheres to simple pythonic code writing and high performance, with an active user and developer community. We choose PyTorch as the backbone DL framework for its accessibility for researchers with different levels of Python proficiency.

**Scikit-Learn**    Scikit-learn [56] is a Python package with extensive machine learning methods. The AUROC curve metric in XBrainLab were implemented through Scikit-Learn built-in functions.

**SciPy**    SciPy [57] is a scientific computing library commonly used for matrices operation or signal processing. XBrainLab time-frequency spectrograms are computed by SciPy's short time fourier transform (STFT) function.

**tkinter**    Tkinter [1] is a lightweight GUI building framework. All GUI features such as interactive widgets, window generation/destruction and actions with callback functions are actualized by tkinter.

**torchinfo**    Torchinfo [2] is a PyTorch supplementary package works similar to the `model.summary()` feature in Tensorflow [58], showing an informative chunk of text contain model structure, size and amount of learnable parameters. A torchinfo object will be generated as soon as a model is evaluated in XBrainLab.

# B    XBrainLab Modules

In this section a more detailed and function-driven description of XBrainLab is provided, with a series of example screenshots.

## B.1    Data import

Supported import file formats XBrainLab includes EEG device output (.edf, .edf+, .gdf and .cnt), EEG toolbox result (.set) and numeric files (.npy, .npz, .mat). The first two types usually contain sufficient information about the recording, and for the numeric files users need to enter the missing

---

[1]Tkinter documention: https://docs.python.org/3/library/tkinter.html
[2]torchinfo GitHub repository: https://github.com/TylerYep/torchinfo

information manually (figure 8b). For all formats alike, subject and session information are often associated with filename, in XBrainLab users can choose to specify the filename template in regular expression to let the software automatically parse subject and session or fill in by hand. To address common scenario that original event markers in file are not in the expression the researchers would like to use in the following analysis, XBrainLab supports loading 1D event label list or 2D event array in MNE expression from .txt or .mat files.

## B.2 Data preprocessing

Preprocessing module in XBrainLab covers basic EEG signal processing such as normalizing, resampling, filtering, normalization, epoching, channel selection and event editing. Each function has a separate pop-up window to take necessary parameters (figure 9, with the current dataset information and preprocess history displayed in the XBrainLab dashboard. Users can reset all steps upon mistakes.

## B.3 Dataset splitting

In experiments, researchers would explore the outcome of different dataset division mimicking real-life EEG applications. The variabilities of EEG could reside in inter- and intra- subjects/sessions/trials/task [59–61], and in XBrainLab we provided subject based, trial based or session based dataset splitting to accommodate user demand. Figure 10 and 11 shows a radical case of splitting dataset into training set and test set by session. Before confirming on the splitting rules, the users can preview the result in a graphical representation (figure 10). After the rules are decided, users can further set the splitting ratio/number/indices, and inspect the contents of splitted dataset (figure 11). All information will be shown in dashboard after splitting operation (figure 11).

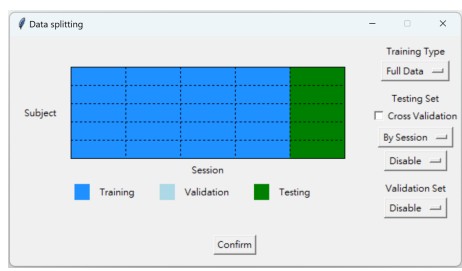

Figure 10: Dataset splitting module in XBrainLab, general split rule setting and preview.

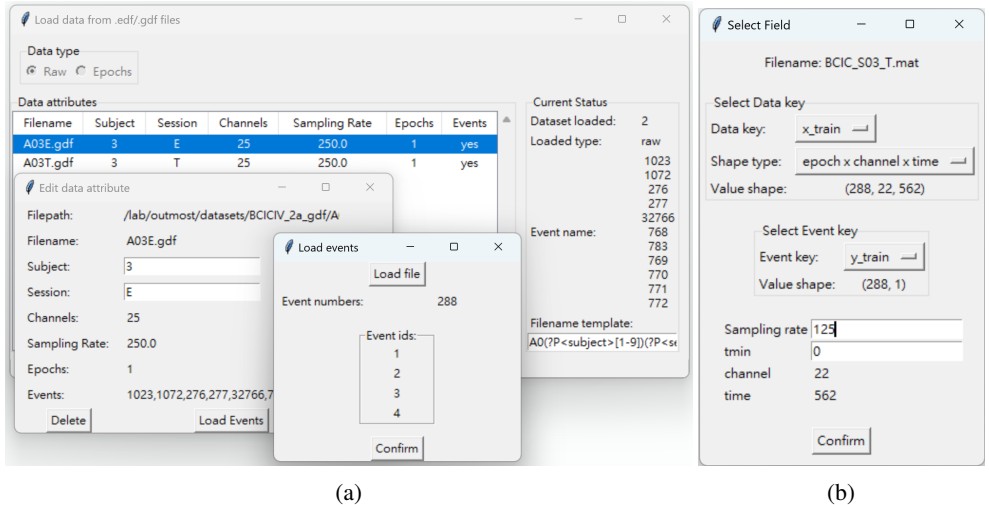

(a)           (b)

Figure 8: Loading module in XBrainLab. (a) From bottom layer: loaded file overview, selected file editing, selected file event loading. (b) Loading window with necessary information entries for numeric files.

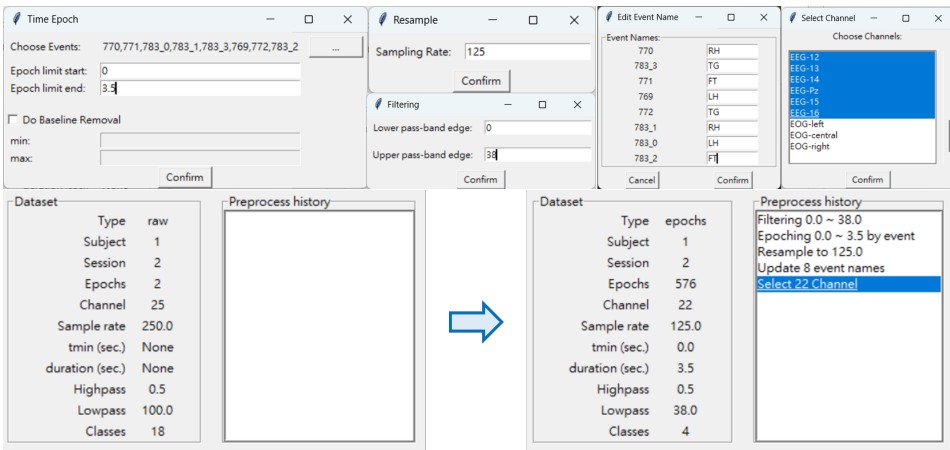

Figure 9: Preprocessing module in XBrainLab. Upper collage: Pop-ups for different preprocessing methods (layout not in operating order for typesetting reasons). Lower collage: data information and preprocess history in XBrainLab dashboard.

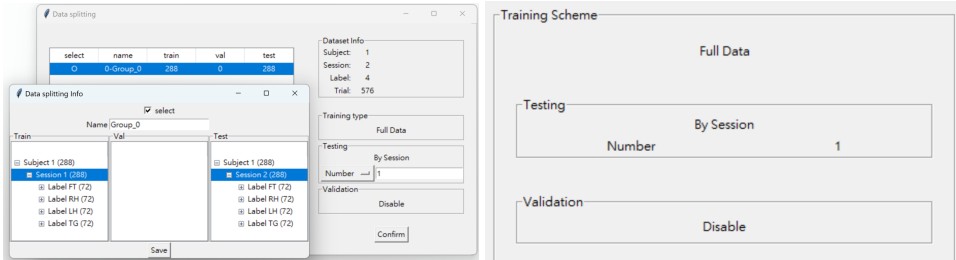

Figure 11: Dataset splitting module in XBrainLab, ratio setting and result inspection.

## B.4  Deep Learning

**Neural networks and training configuration**  XBrainLab has three built-in CNN models, EEGNet, SCCNet and ShallowFBCSPNet. A model instance can be initialized with custom structural parameter such as kernel numbers, or load learned parameters from file (figure 12 upper left). Basic training configurations are aggregated in a setting window (figure 12 upper right), users can fill in the blanks or selected from provided options. The "Checkpoint Epoch" tells how often the user would like to save a model, for example, 25 means in every 25 training epochs, XBrainLab would save current model to the output directory. The "Evaluation" option stands for the criterion to select a model for later visualization and evaluation steps, provided options are: Best validation loss (model with lowest cross-entropy value on validation set), Best test accuracy (model with highest accuracy on test set), Best test AUROC curve score (model with highest AUROC score on test set, AUROC score is computed in one versus rest strategy), Last epoch (model from the last training epoch). "Repeat Number" specifies the times for repeated experiments. XBrainLab provides "Train" mode or "Test Only" mode, in the latter one users can evaluate a model without modifying its hyperparameters. The setting results can be found in the dashboard (figure 12).

**Progress monitor**  During training, training/validation set accuracy, AUROC score and cross-entropy value of each epoch are provided in real-time (figure 13a). After training, the users can observe the trend of accuracy, AUROC score, cross-entropy loss value and learning rate overtime in curve plots (figure 13b). XBrainLab dashboard shows the numbers of finished training epochs; in the "Test Only" mode, the same text frame shows if model inference is complete.

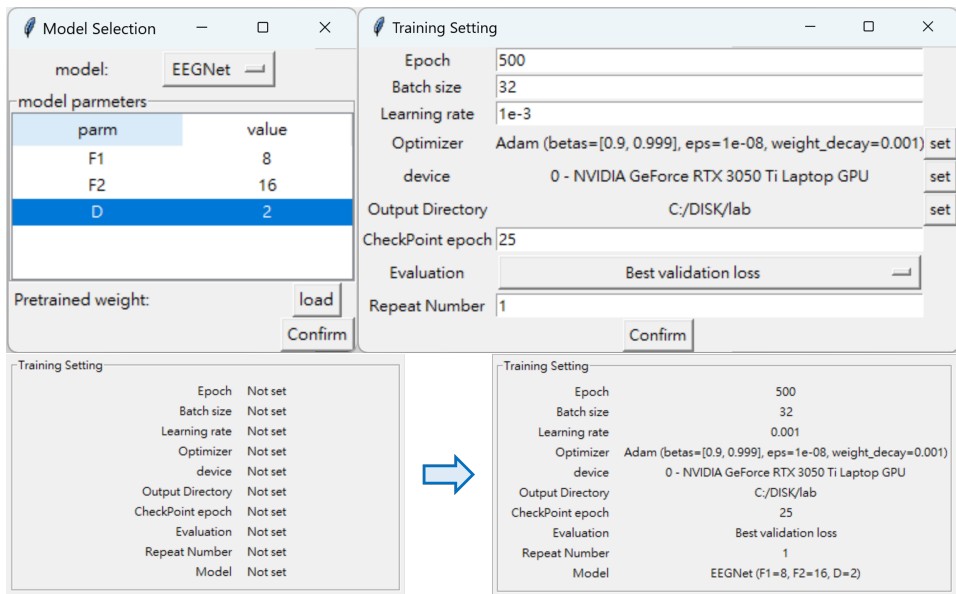

Figure 12: Deep Learning module in XBrainLab, model and training configuration.

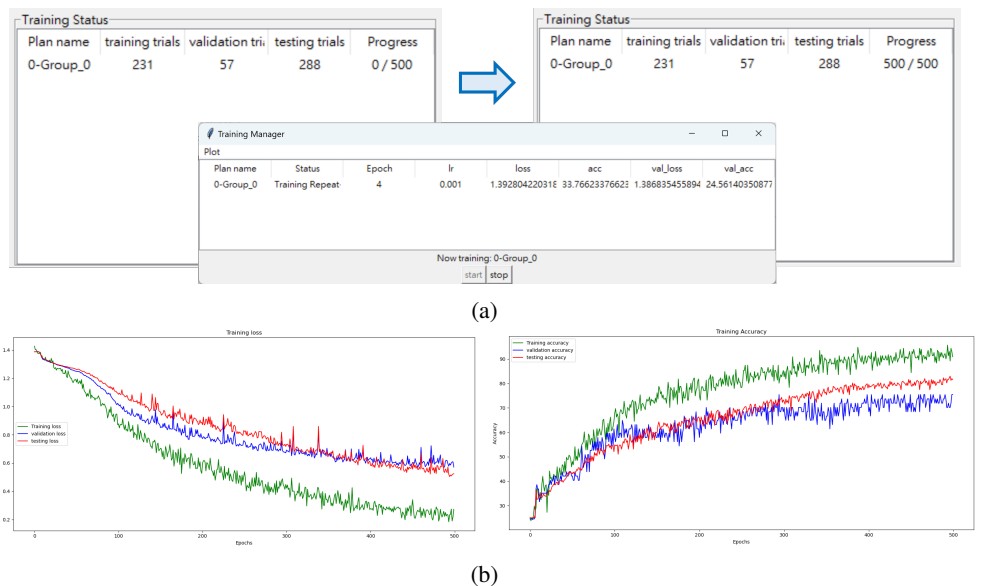

(a)

(b)

Figure 13: Deep Learning module in XBrainLab. (a) Training progress monitor and dashboard status. (b) Learning trends, only loss and accuracy curves are shown here as example.

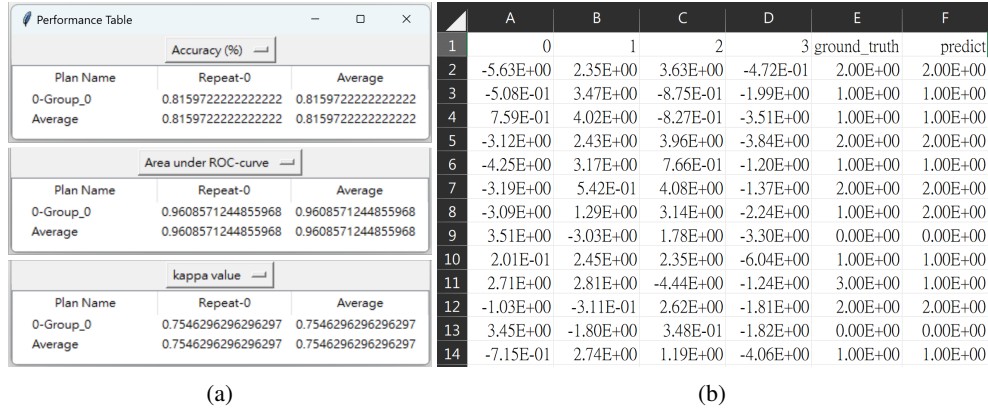

Figure 14: Evaluation module in XBrainLab. (a) Metric tables of all repeats.(b) Exported raw output.

## B.5 Result and analysis

**Model evaluation** After completing "Train" mode model training or "Test Only" mode model evaluating, XBrainLab has obtainable confusion matrix and several metrics: accuracy, AUC score and Cohen's kappa coefficient, from the model satisfying the criterion determined in training configuration setting (Secion A.4) on the test set. The metrics are presented in tables (figure 14a), each row is computed from one model in a repeat, the average over repeats is calculated as well. To enable the researchers to use different metrics to evaluate their model, raw outputs can be exported in .csv format (figure 14b).

**Post-hoc interpretation** The visualization module in XBrainLab currently provides saliency map interpretation plotted in different representations. Saliency map is the gradient values calculated by back-propagating the output of a set of data, commonly believed to show relative importance of each datapoint. The available representations includes "channel-time", "2d topological map" and "3d interactive topological map", with the option to show time domain or frequency domain values as listed in table 2. "Channel-time" could be used to compare the level of activation between time points; "2d topological map" projects a sets of values on 2d electrode positions, and is widely used to examine spatial relations in brain studies; "3d topological map" is projecting values onto a 3d interactive model, which should make the association of electrode position and brain regions more intuitive, and in 3d plotting users can observe the change across time using time selector. More figures can be found in Section 4 in main text and Section B in appendix.

Table 2: XBrainLab Visualization options.

|  | Time domain | | Frequency domain |
|---|---|---|---|
|  | Real | Absolute | Absolute |
| channel-time | ✓ | ✓ | ✓ |
| 2d topomap | ✓ | ✓ | ✓ |
| 3d topomap | ✓ | | |

## B.6 Scripting

The generated script comes in three types: "command", "ui" and "all". "Command" scripts records functional operations and defined parameters when the user was interacting with the GUI, "ui" scripts is independent from XBrainLab functions and records where new plotting windows were opened, and "all" is the combined script of the previous two. To show XBrainLab GUI dashboard after running a "command" script, the user needs to add `XBrainLab_instance.show_ui()` to the last line, from which the new GUI will be automatically filled with operations done in the code script.

## B.7 Custom extensions

The code structure of XBrainLab are innately designed for easy modification with custom add-ons or plugins. As long as the new script takes/provides matching arguments to the ones in the same hierarchy, the master modules can automatically accustom the added functions. Current limitation

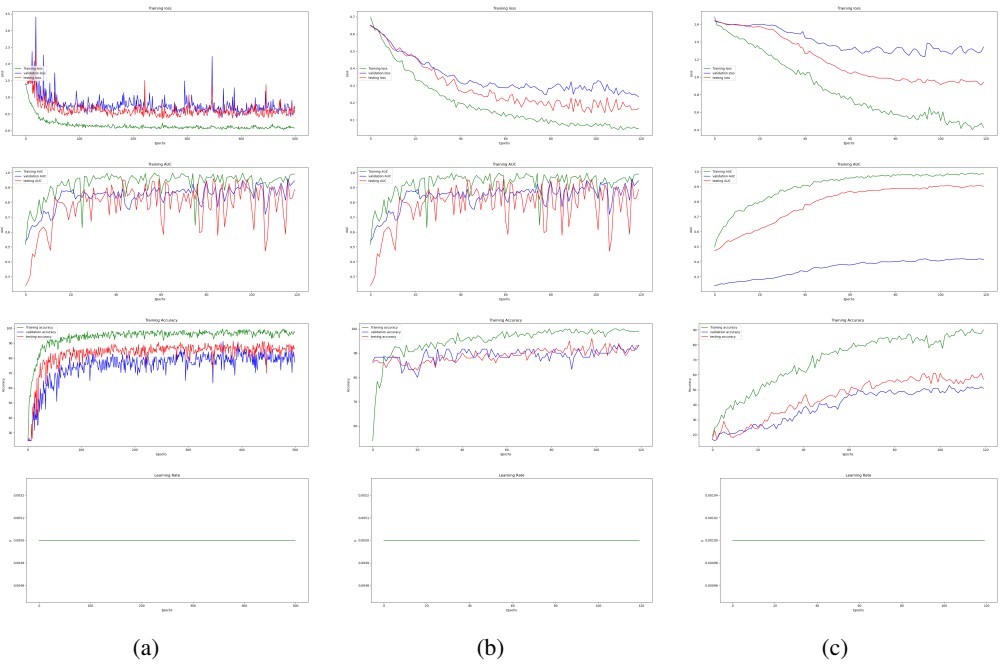

(a)                        (b)                        (c)

Figure 15: Training monitor over all training epochs from the case studies. Row 1: cross-entropy loss. Row 2: AUC score. Row 3: accuracy. Row 4: learing rate. (a) MI dataset. (b) ERN dataset. (c) SSVEP dataset.

is seating on that the user still needs to write their own codes for the GUI when adding a more fundamental module and wish to reach the added functions through the dashboard.

## C   Additional figures of case studies

In this section we provide the exhibition and discussion about additional graphical results from the case studies.

**Training progress monitor**    Figure 15 illustrates the loss, AUROC score, accuracy and learning rate of the case studies. The general trend of dropping loss value, increasing accuracy/AUROC curve can be viewed as the model did learn about the feature distribution in the training data. Our case studies were set with static learning rates. Notably, the values for validation set and test set in each epoch is computed after updating the hyperparameters once with the training set, hence the values can not be read as "random guess" result by the model at the beginning of a training round.

**Motor Imagery**    Figure 16 supplements the case study on MI dataset. Fig. 16b and 16a are saliency maps with and without taking absolute values, from these figures we obtain similar inference as in section 4.1 regarding the MI spatial features. The faint concentration of extreme values on the time samples does not bear much resemblance to SOTA DL model interpretation results [39] even if the time scale is aligned. Nevertheless, the DL model and data segments are not identical, it'll be safer to say that the model in analysis may have deemed the densely colored parts are relatively more important toward the model's final decision in the inference data set. For time-frequency spectrogram topological representation (figure 16d), the figure only states frequency pattern exist in some electrodes; spectrogram in time-frequency representation harboured more commonly acknowledged information such as prominent EEG rhythms and their surfacing times, thus it's suggested to take both representations into account simultaneously when reasoning with the visualization results.

**Event Related Potential**    The confusion matrix of our case study on ERN dataset is unsurprisingly biased 17, showing classification result leaning into "correct" due to severe class imbalance in

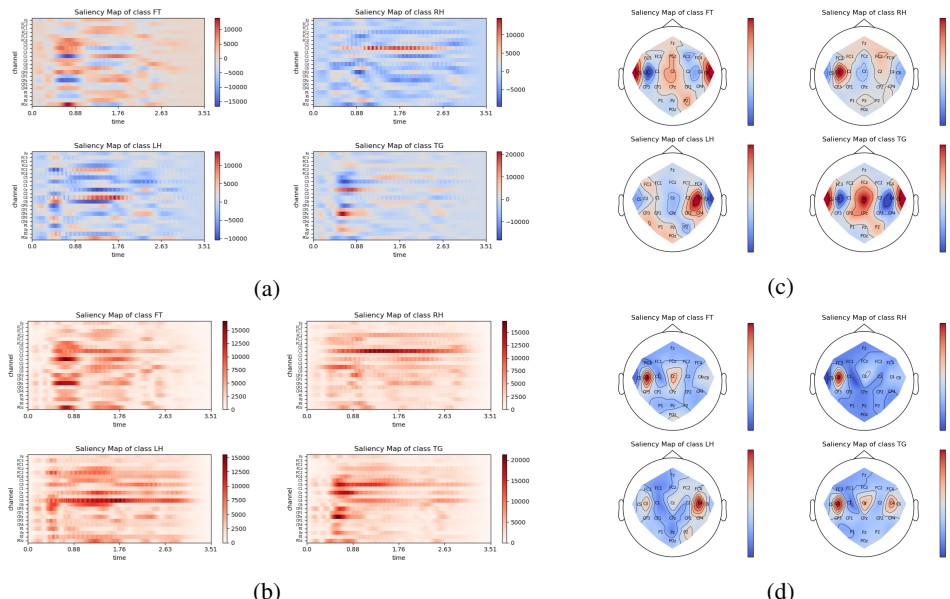

Figure 16: Additional interpretations of MI dataset. (a) Saliency map without taking absolute values. (b) Saliency map taking absolute values. (c) Saliency topological map without taking absolute values. (d) Saliency map time-frequency spectrogram value of each electrode averaging over trials and timesamples, plotted as topological map.

the dataset [39]. Figure 18 supplements the case study on ERN dataset, the additional model visualizations support the discussion on temporal and spatial activations in section 4.2. The time-frequency spectrogram shows pattern in low frequencies, which is in line with the time-frequency spectrograms of the original data plotted in literature [62].

**Steady State Visual Evoked Potential** Figure 19 supplements the case study on SSVEP dataset. A complete set of topological map in 2d and 3d are exhibited here, directly showing the previous observations on saliency map features and their association with occipital lobe.

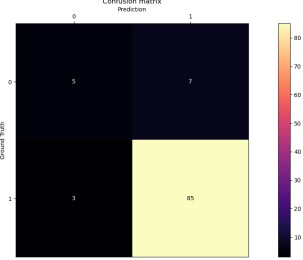

Figure 17: Confusion matrix on ERN dataset.

# D    Limitation

Errors occur by happenstance when executing XBrainLab on some machines with macOS. It's currently unclear this problem is induced by specific type of processor, version of operating system or other factors. However, in our own tests and private release of earlier versions to controlled users, XBrainLab runs smoothly on Windows and Linux.

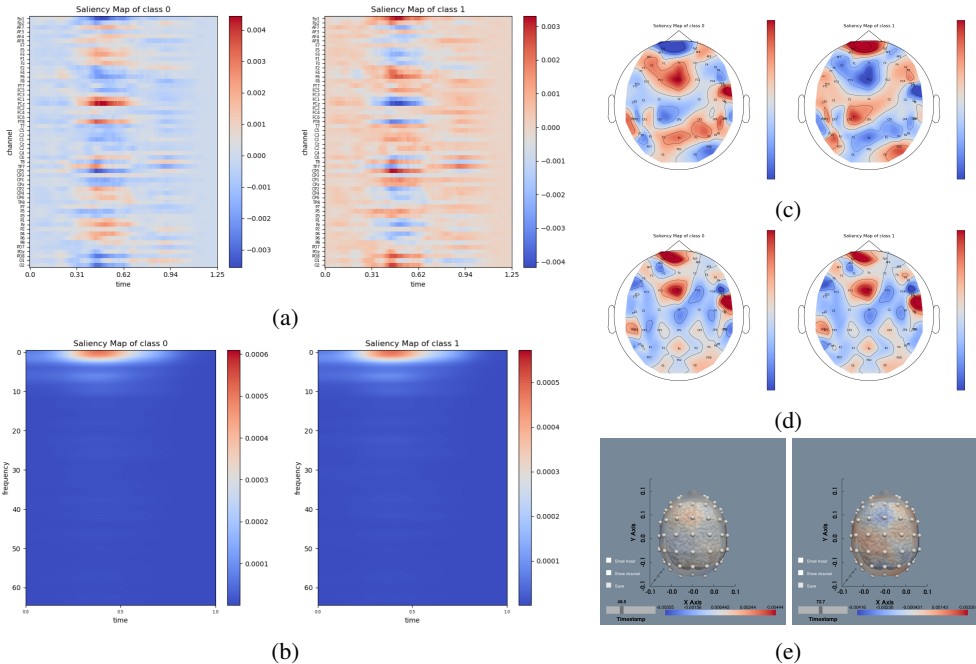

Figure 18: Additional interpretations of ERN dataset. (a)

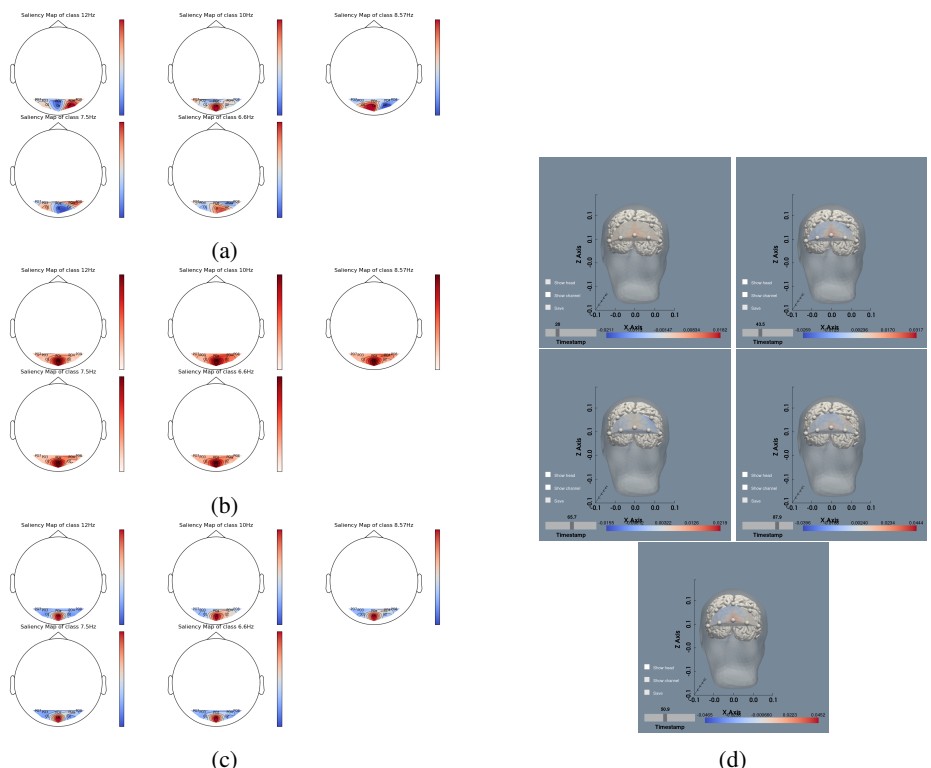

Figure 19: Additional interpretations of SSVEP dataset. (a)

