# OpenReview forum: "XBrainLab: An Open-Source Software for Explainable Artificial Intelligence-Based EEG Analysis"
_NeurIPS.cc/2023/Workshop/AI4Science — NeurIPS2023-AI4Science Poster_

### Official Review · Reviewer_utAA · 2023-10-20
**The package XBrainLab seems to be a useful, no-code solution, but the only novel contribution is an in-built interpretability analysis (confusion matrix/saliency maps) which are common metrics in any model training pipeline.**

**Rating:** 5
**Confidence:** 4

**Review:**

The authors introduce a low/no-code package to train CNNs that decode EEG brain signals and visualize results in a GUI. The results appear to align with what neurology experts would empirically observe. The package splits the decoding pipeline into data loading, preprocessing, training and interpretation. The interpretation consists of saliency maps and confusion matrices, which are common metrics in various Machine Learning and deep learning pipelines. The automatic generation of these auxillary figures and latent feature analysis is valuable. However, there is limited discussion on the choice, accuracy or evaluation of the 3 CNN models and how they compare to more recent/SOTA methods.

XBrainLab appears to be well designed and would be useful for EEG researchers in practice, but this work itself provides little novel contribution to the field. The architectures of trainable models are adopted from existing literature, and the interpretability metrics are common practice when training models in independent pipelines.

---

### Official Review · Reviewer_R6N4 · 2023-10-25
**XBrainLab**

**Rating:** 10
**Confidence:** 5

**Review:**

The authors present an amazing python-based tool for EEG analysis that builds on the decades on work on signal processing and analysis. Authors not only present their own work, but they also show a comparative account of all avaialble open source EEG signal analysis tools. The complete pipeline of events in the processing of EEG data has been considered while designing the interface and it is clearly evident in the dashboard view. Furthermore, they have GUI scripting and explainable deep learning. I am elated to read this paper and cannot wait to try this tool for my own research.

---

### Meta-Review · Area_Chair_jhAs · 2023-10-26

**Recommendation:** Accept (Poster)
**Confidence:** 2

**Metareview:**

The reviewers identify the main contribution of this work as the visualization software platform rather than an AI methodological contribution for Science. While potentially useful, it is not very clear what additional functionality this particular visualizer provides over existing software: particularly braindecode and EPViz (https://github.com/jcraley/epviz), developed specifically for EEG and allowing integration of DL model predictions.

Given the split in the reviewer decision, I would recommend this paper be presented as an accepted poster and suggest additional discussion be included to make the contributions more concrete.